# Spectrophotometric Determination of the Aggregation Activity of Platelets in Platelet-Rich Plasma for Better Quality Control

**DOI:** 10.3390/dj7020061

**Published:** 2019-06-03

**Authors:** Tetsuhiro Tsujino, Kazushige Isobe, Hideo Kawabata, Hachidai Aizawa, Sadahiro Yamaguchi, Yutaka Kitamura, Hideo Masuki, Taisuke Watanabe, Hajime Okudera, Koh Nakata, Tomoyuki Kawase

**Affiliations:** 1Tokyo Plastic Dental Society, Kita-ku, Tokyo 114-0002, Japan; tetsudds@gmail.com (T.T.); kaz-iso@tc4.so-net.ne.jp (K.I.); hidei@eos.ocn.ne.jp (H.K.); sarusaru@mx6.mesh.ne.jp (H.A.); y-sada@mwd.biglobe.ne.jp (S.Y.); shinshu-osic@mbn.nifty.com (Y.K.); hideomasuki@gmail.com (H.M.); watatai@mui.biglobe.ne.jp (T.W.); okudera@carrot.ocn.ne.jp (H.O.); 2Bioscience Medical Research Center, Niigata University Medical and Dental Hospital, Niigata 951-8520, Japan; radical@med.niigata-u.ac.jp; 3Division of Oral Bioengineering, Institute of Medicine and Dentistry, Niigata University, Niigata 951-8514, Japan

**Keywords:** platelet-rich plasma, platelets, aggregation, spectrophotometer, quality assurance

## Abstract

Although platelet-rich plasma (PRP) is now widely used in regenerative medicine and dentistry, contradictory clinical outcomes have often been obtained. To minimize such differences and to obtain high quality evidence from clinical studies, the PRP preparation protocol needs to be standardized. In addition, emphasis must be placed on quality control. Following our previous spectrophotometric method of platelet counting, in this study, another simple and convenient spectrophotometric method to determine platelet aggregation activity has been developed. Citrated blood samples were collected from healthy donors and used. After centrifugation twice, platelets were suspended in phosphate buffered saline (PBS) and adenosine diphosphate (ADP)-induced aggregation was determined using a spectrophotometer at 615 nm. For validation, platelets pretreated with aspirin, an antiplatelet agent, or hydrogen peroxide (H_2_O_2_), an oxidative stress-inducing agent, were also analyzed. Optimal platelet concentration, assay buffer solution, and representative time point for determination of aggregation were found to be 50–100 × 10^4^/μL, PBS, and 3 min after stimulation, respectively. Suppressed or injured platelets showed a significantly lower aggregation response to ADP. Therefore, it suggests that this spectrophotometric method may be useful in quick chair-side evaluation of individual PRP quality.

## 1. Introduction

Since the first report by Marx et al. [1], regenerative therapy using platelet-rich plasma (PRP) has been increasingly used as a promising therapeutic method for more than two decades. However, in several tissues, such as bone, PRP therapy has not produced positive clinical outcomes. Differences in bone regeneration have been thought to mainly be because of individual differences in blood samples. Few efforts to overcome this hurdle have been made at the national and international levels. As a result, clear and strong evidence on the use of PRP, which can be adopted by individual national regulatory agencies and thereby support the clinical use of PRP, has not yet been obtained [2].

It could be argued that such differences could be eliminated by increasing the sample size. However, in the case of home-made PRP, prepared at the time of use, quality cannot be controlled or assured like factory-made products. Differences in individual PRP preparations are also due to the different preparation protocols, devices, and operator skills. Without careful consideration of the mentioned technical biases, reliable randomized clinical trials cannot be initiated for PRP therapy. Despite this situation, a recent proposal to standardize the PRP preparation protocol is expected to improve the quality of clinical evidence. Furthermore, sharing the concept of “quality control” among individual clinicians will complement the current standardization movement to assure similarity in the quality of individual PRP preparations in the near future.

In general, the quality of the cell-based medicinal products (CBMPs) is defined and evaluated mainly based on the following five parameters: sterility, purity, identity, potency, and stability [3]. However, home-made PRP is distinguished from the typical CBMP, such as hematopoietic stem cells, due to various factors. PRP quality can be defined by platelet count, growth factors present and their levels, coagulation activity, and platelet activity. Contamination with leukocytes might be an additional parameter influencing the quality [4,5]. It is not easy to determine platelet and leukocyte counts without using an automated hematology analyzer, which cannot be easily installed in dental clinics due to high initial investment and space required. Therefore, in a previous study [6], a simple and convenient assay method to quickly determine the platelet count using a pocketable spectrophotometer was developed. Coagulation activity is routinely determined by pocketable machines, e.g., CoaguChek^®^ XS Plus (Roche, Basel, Switzerland). Thus, assays to rapidly determine the growth factor levels and platelet activity need to be developed.

This study focuses on platelet aggregation activity as a representative of platelet function. Moreover, it focuses on a microplate-reader-based technology to evaluate platelet aggregation as an alternative to the conventional type of aggregometer [7,8,9,10]. According to this principle, we have developed a spectrophotometric assay to evaluate adenosine diphosphate (ADP)-induced platelet aggregation in a quantitative manner. The primary purpose of this study was to test our modified spectrophotometric assay for platelet aggregation. The secondary purpose was to validate this assay method to evaluate the quality of platelets contained in individual PRP preparations by establishing the reference range and improving the quality of clinical evidence in cooperation with other assay methods. We successfully validated the assay’s applicability by comparing normal platelets with suppressed and injured platelets. Although this study was limited by the size and variation of samples, we successfully optimized the assay conditions and suggest its applicability in quality control of individual PRP preparations.

## 2. Materials and Methods

### 2.1. Preparation of P-PRP and Platelet Suspension

Blood samples were collected from 14 healthy, nonsmoking adult volunteers in the age group of 22 to 70 (Male: N = 11, mean = 51.5 y, Female: N = 3, mean = 25.3 y) using butterfly needles (21G × ¾in.; NIPRO, Osaka, Japan). Despite having lifestyle-related diseases and taking medication, these donors had no limitations on the activities of daily living. These donors also declared to be free of Human immunodeficiency virus (HIV), Hepatitis B virus (HBV), Hepatitis C virus (HCV), or syphilis infections. In addition, a prothrombin test was performed on all the blood samples by means of CoaguChek^®^ XS, and all the samples were found to be normal in blood cell counts.

Peripheral blood samples (~7 mL) were collected into plastic vacuum plain blood collection tubes (Neotube; NIPRO, Osaka, Japan) containing 1 mL of acid–citrate–dextrose formula A (ACD-A; Terumo, Tokyo, Japan). Fresh whole-blood samples were immediately centrifuged at 530× *g* for 10 min or stored for up to 2 days before centrifugation at ambient temperature [11,12]. The upper plasma fraction, known as the platelet-rich plasma (PRP) fraction, was collected, transferred into fresh sample tubes, and treated for 10 min with 1 μg/mL prostaglandin E_1_ (PGE_1_; Cayman Chemical, Ann Arbor, MI, USA) [13,14]. The PRP fraction was again centrifuged at 2650× *g* for 3 min. Precipitated platelets were gently resuspended in acellular plasma, Tyrode buffer solution, phosphate buffered saline (PBS) or PBS containing ethylenediaminetetraacetic acid (EDTA) (final concentration: approximately 1.5 mg/mL). The platelet suspension in acellular plasma was designated as “pure PRP (P-PRP).” The number of platelets and other blood cells within the whole-blood samples and platelet suspensions was determined using an automated hematology analyzer (pocH 100iV, Sysmex, Kobe, Japan).

The study design and consent forms for all the procedures (project identification code: 2297) were approved by the Ethics Committee for Human Subjects of the Niigata University School of Medicine (Niigata, Japan) on 14 October 2015, in accordance with the Helsinki Declaration of 1964 as revised in 2013.

### 2.2. Spectrophotometric Determination of Platelet Aggregation

P-PRP and other platelet suspensions were serially diluted with equal volume of acellular plasma or the corresponding buffer solutions. Series of diluted platelet suspensions were measured using a pocketable spectrophotometer (PiCOSCOPE, Ushio Inc., Tokyo, Japan) [6]. The spectrophotometer can be operated by remote control through a specific application installed on smart devices, including the iPad Air (Apple, Cupertino, CA, USA). Platelet suspensions were transferred into 0.2 mL highly transparent PCR tubes (Nippon Genetics Co., Ltd., Tokyo, Japan) and treated with 5 μM ADP (Wako Pure Chemicals, Osaka, Japan).

To prepare dysfunctional platelet models, we pretreated P-PRP with 0.1 mg/mL aspirin (acetylsalicylic acid; Wako Pure Chemicals, Osaka, Japan) or 10 μM H_2_O_2_ (Wako) for 30 min at 22–24 °C. The absorbance was measured at an interval of one minute at 615 nm (range of wavelength: 570–660 nm). At the end of measurement, each blank was measured as the absorbance of 100% aggregation.

### 2.3. Statistical Analysis

Data were expressed as mean ± standard deviation (SD) or box plot (Figure 1). For multigroup comparisons, statistical analyses were performed to compare the mean values by Kruskal–Wallis one-way analysis of variance, followed by Steel–Dwass multiple comparison test (BellCurve for Excel; Social Survey Research Information Co., Ltd., Tokyo, Japan). Differences with *p* < 0.05 were considered statistically significant.

## 3. Results

### 3.1. Effect of Different Assay Buffer Solutions, Time Points, and Platelet Densities on the Assay System

The effect of different assay buffer solutions on the ADP-induced platelet aggregation assessed by the assay system at 3 min is shown in Figure 1. For platelets suspended in acellular plasma, the percentage inhibition levels were lower than those of others, based on the platelet density tested. However, the effects of ADP were sustained at lower platelet densities for platelets suspended in the Tyrode buffer solution. In contrast, for the platelets suspended in PBS and EDTA-containing PBS, the effects of ADP were constantly sustained between approximately 15%–20%, which was in the range of the platelet densities tested.

The effect of platelet density on ADP-induced platelet aggregation over a time course as assessed by the assay system is shown in Figure 2. Generally, platelets suspended in Tyrode buffer solution showed a better response to ADP because of the presence of Ca^2+^; however, as the platelet density increased, the platelets tended to aggregate in this solution. In about a half of the samples, we found macroscopically identifiable platelet aggregates immediately after suspension in this solution. We did not use these suspensions for data collection, since the platelet number could not be counted. In contrast, platelets in PBS and EDTA-containing PBS responded similarly to ADP, although an increase in the platelet density slightly reduced platelet responsiveness (i.e., percentage inhibition).

### 3.2. Effect of Different Platelet Conditions on the Assay System

The effect of platelet condition on ADP-induced platelet aggregation assessed by the assay system at 3 min is shown in Figure 3. Except at lower platelet densities, the base line was stable and sustained at similar levels. Response to ADP was significantly reduced in platelets suppressed by aspirin and injured by H_2_O_2_.

The effect of platelet density on ADP-induced aggregation of dysfunctional platelets over a time course assessed by the assay system is shown in Figure 4. Regardless of platelet density, both aspirin and H_2_O_2_ significantly reduced platelet responsiveness to ADP. However, it should be noted that at a higher platelet density, the degree of reduction apparently decreased.

## 4. Discussion

In this study, we used blood samples collected from healthy donors who did not receive any medication and stored them for up to 2 days prior to preparation of pure PRP (P-PRP). After double centrifugation, platelets were resuspended in PBS and stimulated with ADP. Our spectrophotometric assay method demonstrated that these platelets responded to ADP similarly regardless of individual differences. After 3 min of stimulation, although the platelet count slightly increased, the absorbance (raw value) decreased by 10%–20% in response to ADP. Moreover, we also found that platelets treated with aspirin or H_2_O_2_ showed significantly reduced responsiveness to ADP.

These observations are further analyzed in detail below. To optimize the analysis conditions, we focused on (1) assay buffer solutions, (2) a range of platelet densities and (3) end points for data collection. In addition, to validate this method, we developed and examined (4) models of partially dysfunctional platelets using aspirin and H_2_O_2_.

### 4.1. Optimal Assay Buffer Solutions

In a previous study [6], based on handling efficiency, we chose acellular plasma to prepare platelet suspensions, i.e., P-PRP, and validated the applicability of the spectrophotometric method to determine platelet count. The plasma was a good “buffer solution” that could be used to easily and efficiently suspend precipitated platelets. However, in this study, we showed that although PRP is the recommended type of platelet suspension for aggregometric assays [15], plasma substantially reduced ADP-induced platelet aggregation. This is probably due to ectonucleotidases and/or similar enzymes present in the plasma, which quickly degrade ADP. Alternatively, certain plasma proteins, such as albumin, may block contact between ADP and its specific platelet receptors.

It is known that Ca^2+^ plays a crucial role in maintaining platelet functions, such as aggregation. Thus, we analyzed the effects of the Ca^2+^ and Mg^2+^-containing Tyrode buffer solution as an alternative to plasma, because this buffer has been frequently used in the conventional assay involving a platelet aggregometer. However, the Tyrode buffer induced platelet aggregation at high rates in our experimental system, especially in the case of freshly prepared P-PRP, even though prostaglandin E_1_ (PGE_1_) was added to P-PRP prior to suspending platelets in the Tyrode buffer. Thus, we chose PBS to investigate the Ca^2+^-induced platelet activation and morphological changes, as in our previous studies [16,17].

We then examined Ca^2+^, and Mg^2+^-free PBS and added EDTA to further eliminate the divalent cations. In general, PBS enabled suspension of precipitated platelets more efficiently than Tyrode buffer solution. In case of freshly prepared P-PRP, addition of EDTA caused efficient suspension of platelets in PBS without forming aggregates and sacrificing platelet responsiveness. This was confirmed by macroscopic examination and platelet counts. However, the superior effects of EDTA were not observed for blood samples stored over 2 h.

Also considering its low cost, high efficiency, and easy availability, we concluded that PBS can be used in a chair-side spectrophotometric assay for platelet aggregation.

### 4.2. Optimal Range of Platelet Density

Light transmission was examined through the bottom of the PCR sample tubes in this spectrophotometer. When platelet density was relatively low, the corresponding absorbance tended to increase gradually with gravity-dependent platelet sedimentation. This tendency could be counteracted by increased platelet density in a limited period of time. In contrast, when platelet densities were relatively high, platelet aggregation occurred, and no increase in absorbance was seen. This phenomenon may lead to underestimation of platelet activity. Therefore, we concluded that the optimal range of platelet density is roughly 50–100 × 10^4^/μL.

### 4.3. Optimal End Point

Aggregation of human platelets in vitro can occur in two phases—primary and secondary aggregation [18]. With the pocketable spectrophotometer, we could not monitor changes in absorbance continuously; thus, we could not identify the pattern of aggregation through our assay system. The change in absorbance usually plateaued within 1–2 min and apparent additional changes did not occur within 5 min after stimulation. Therefore, any time point after 1 min of stimulation may be acceptable as an end point. Considering handling efficiency and probability of misoperation, we chose 3 min as an end point.

### 4.4. Comparisons with Suppressed or Injured Platelets

Under the mentioned optimal conditions, normal platelets were compared with artificially modified platelets. Aspirin is a conventional antiplatelet agent that inhibits cyclooxygenase-1, which forms thromboxane A_2_ in response to secreted dense granule constituents, such as serotonin and ADP, in the secondary wave [18]. Thus, platelets treated with aspirin were expected to show reduced aggregation activity in response to ADP. We observed only approximately 50% inhibition when P-PRP was treated with 0.1 mg/mL aspirin for 20 min. Modification of the timing (from P-PRP to platelet suspension in PBS), duration (from 20 min to overnight), and concentration (from 0.1 mg/mL to 0.3 mg/mL) of aspirin did not significantly augment the inhibition. However, these findings indicate that the assay method is capable of distinguishing suppressed platelets from normal platelets.

Endogenously generated H_2_O_2_ is thought to function as a trigger for platelet activation and aggregation [19,20]. On the contrary, exogenously added H_2_O_2_, which is known as an inducer of oxidative stress, damages the plasma membrane through lipid peroxidation and alters fluidity and leakiness of the membrane. As a result, H_2_O_2_ inhibits ADP-dependent platelet activation [21]. In this study, as observed with aspirin, H_2_O_2_ inhibited ADP-induced platelet aggregation by approximately 50%. Thus, this suggests that when platelets are damaged by manual error during preparation or donor-dependent oxidative stress, this assay method is capable of indicating reduced PRP quality.

### 4.5. Limitations of This Study

Platelet aggregation activity assessed by conventional aggregometry is expressed by qualitative or semi-qualitative data. Unlike the platelet count, such data cannot be easily used for comparison with data from other samples. Moreover, Chandrashekar [22] mentioned that light transmission aggregometry lacks standardization and normal reference values are not widely available. However, our findings indicate the possibility that this simple and quick assay method could be used to assure individual PRP quality. At the same time, we should consider the limitations of this study, which could cause biases that lead to misinterpretation of the data.

ADP is one of the most important mediators of hemostasis and thrombosis, and thus has been used in conventional clinical laboratory testing of platelet function [23]. In addition, because of its simple chemical constitution, we chose ADP for the development of the assay method. The concern regarding the suitability of the aggregation assay in the assessment of PRP quality can be explained as follows. Since clot formation of PRP is induced by coagulation factors, such as thrombin or CaCl_2_, or contact with glass surface in the presence of Ca^2+^, it is obvious that platelets are activated primarily by thrombin and fibrin, but not by ADP. However, growth factor secretion, which is related to the most important criterion of PRP quality, and aggregation are simultaneous coupling events of activated platelets. Therefore, we thought that even with variations in the manners and/or degrees of platelet activation with individual meditators, ADP is acceptable for preparation of the basic model of activated platelets.

On the other hand, this study was done using a small sample size consisting of volunteers of Japanese ethnicity. Therefore, individual differences are relatively lower and we may have obtained statistical significances in ADP-induced aggregation between control platelets and the dysfunctional ones. However, recent advances in platelet studies have revealed that platelet functions, which include responses to ADP and sensitivity to aspirin, vary qualitatively and quantitatively with genetic differences among races and individuals [23,24,25,26]. To efficiently detect inherited platelet disorders or dysfunctional platelets by reducing possible quantitative variations, we performed the assay with ADP at a concentration higher than the endogenous levels and aspirin at a concentration higher than the therapeutic levels. We concluded that these choices were optimal for our samples.

However, to realize this assay method as a globally standardized protocol for assessment of PRP quality in regenerative dentistry, further verification and subsequent improvement should be done with larger sample sizes by organizing international round robin testing. In addition, to evaluate PRP quality comprehensively in a timely manner, this kind of platelet function testing should be done with other tests regarding growth factor levels, platelet counts, and coagulation activity.

## 5. Conclusions

Our assay method, developed to evaluate platelet aggregation activity, is simple, quick, and sensitive enough to detect dysfunctional platelets. In addition, this method does not require high initial investment, technical training, or space in clinics. At present, we have no evidence that platelet aggregation activity influences PRP clinical efficiency; however, this assay method, in combination with other assay methods and standardization of preparation protocols, will enable assurance of individual PRP quality and facilitate high quality randomized controlled trials to obtain strong evidence in support of PRP therapy. This will further aid in terminating the current endless debate on PRP efficiency.

## Figures and Tables

**Figure 1 dentistry-07-00061-f001:**
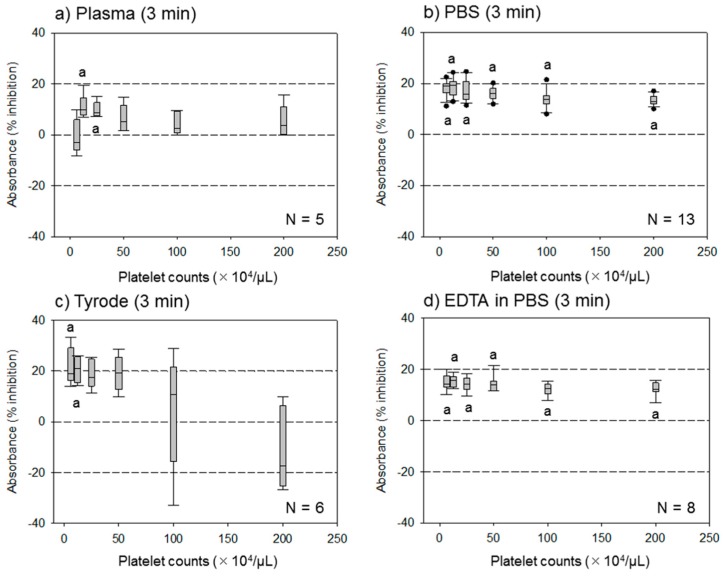
Effects of different assay buffer solutions on the adenosine diphosphate (ADP)-induced platelet aggregation. Platelets were suspended in (**a**) acellular plasma, (**b**) phosphate buffered saline (PBS), (**c**) Tyrode buffer solution, or (**d**) ethylenediaminetetraacetic acid (EDTA)-containing PBS at the indicated densities and stimulated with 5 µM ADP for 3 min at 22–24 °C. ^a^
*p* < 0.05 compared with individual corresponding controls at 0 min.

**Figure 2 dentistry-07-00061-f002:**
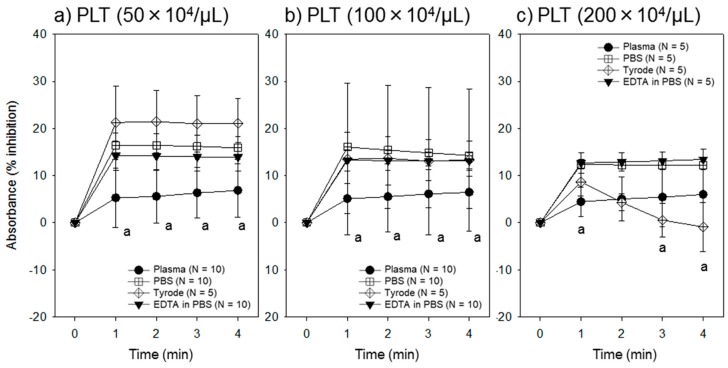
Effect of platelet density on ADP-induced platelet aggregation over a time course. Platelets were suspended in acellular plasma, PBS, Tyrode buffer solution, or EDTA-containing PBS at a density of (**a**) 50 × 10^4^/µL, (**b**) 100 × 10^4^/µL or (**c**) 200 × 10^4^/µL and stimulated with 5 µM ADP for up to 4 min at 22–24 °C. ^a^
*p* < 0.05 compared with the data of PBS at the same time points.

**Figure 3 dentistry-07-00061-f003:**
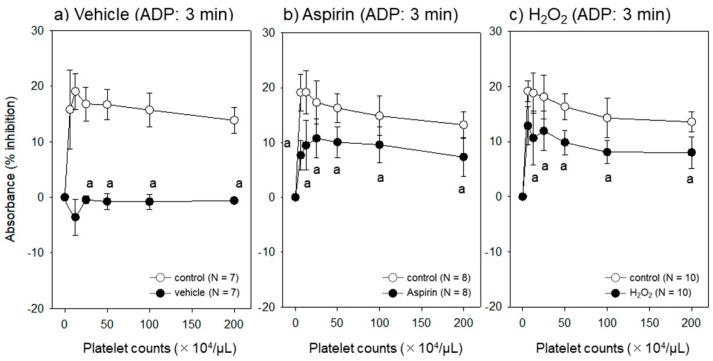
Effect of different platelet conditions on the ADP-induced platelet aggregations. Platelets suspended in acellular plasma were treated with (**a**) a vehicle of aspirin (0.1% dimethyl sulfoxide), (**b**) aspirin or (**c**) H_2_O_2_ for 30 min prior to resuspension in PBS and were stimulated with 5 µM ADP for 3 min. (**a**) The base line was monitored with no addition. ^a^
*p* < 0.05 compared with the control at the same platelet densities.

**Figure 4 dentistry-07-00061-f004:**
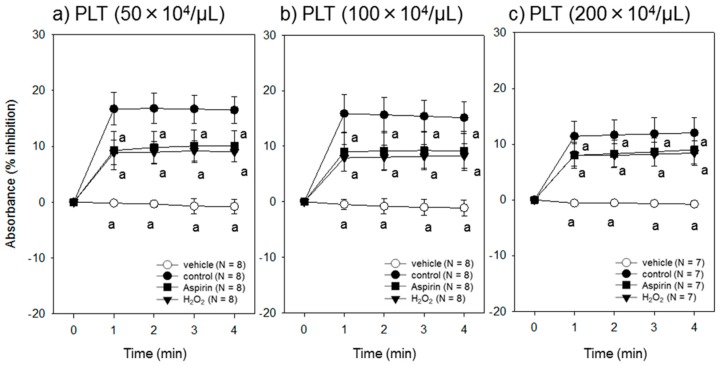
Effect of platelet densities on ADP-induced aggregation of dysfunctional platelets over a time course. Platelets were treated with a vehicle of aspirin (0.1% dimethyl sulfoxide), aspirin or H_2_O_2_, resuspended in PBS at a density of (**a**) 50 × 10^4^/µL, (**b**) 100 × 10^4^/µL or (**c**) 200 × 10^4^/µL and stimulated with 5 µM ADP for up to 4 min at ambient temperature. ^a^
*p* < 0.05 compared with the corresponding controls at the same time points.

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
