# Peer review of "Spectrophotometric Determination of the Aggregation Activity of Platelets in Platelet-Rich Plasma for Better Quality Control"

_dentistry, 2019, doi:10.3390/dj7020061_

Round 1
Reviewer 1 Report
Some English language and style corrections are required
The article focuses on simple, bedside method for platelet aggregation measurement. Although PRP is getting wider interest in the treatment of musculoskeletal disorders there is unsolved question which PRP is optimal? Although there is no studies regarding PLT aggregation and its clinical efficacy there papers aims to establish easy to carry method, that may in the future be a "holy gral". The study is carried in well organized manner, the results are clear, but more studies are necessary for establishing reproducible method, and even more work is necessary to asses it's clinical value. I believe this study may benefit PRP studies in the future.
Author Response
Thank you for your evaluation. The revision was done with the assistance of a native English speaking medical editor contracted to a professional editing service.
The purpose of this study was to validate our modified spectrophotometer-based assay in screening the quality of platelets contained in individual PRP preparations. For this purpose, it is much better to collect samples from many donors with various systemic health conditions. However, it was difficult for us to test it with large sample sizes. Instead, we artificially prepared dysfunctional platelets using aspirin and hydrogen peroxidase. As expected, we successfully validated that our method is simple but sensitive enough to determine the aggregation activity of platelets.
As you indicated, we also think that we need to test many more samples and improve the methods to be more widely applicable in quality assurance of PRP preparations prior to clinical study.
Reviewer 2 Report
The manuscript „Spectrophotometric determination of the aggregation activity of platelets in platelet-rich plasma for better quality control“ is scientfic paper. This study focuses on platelet aggregation activity and assay method to evaluate activity. This manuscript contribute to find simple, low cost and effective method applicable in clinical work.
The reference should be written according to the instructions.
Author Response
Thank you for your evaluation. We checked and modified the reference section to meet the journal style.
Reviewer 3 Report
The subject of study is very interesting; However, authors should order and organize the paper
I should add clearly and precisely the objective of the study
The authors indicate the possibility that this simple and quick assay method could be used to assure individual PRP quality. The method, developed to evaluate platelet aggregation activity, is simple, quick, and sensitive enough to detect dysfunctional platelets. In addition, this method does not require high initial investment, technical training, or space in clinics. PRP quality and facilitate high quality randomized controlled trials to obtain strong evidence in support of PRP therapy.
Introduction
State specific objectives, including any prespecified hypotheses
Preparation of P-PRP and platelet suspension Give the eligibility criteria, and the sources and methods of selection of participants
The discussion should be modified Discuss limitations of the study, taking into account sources of potential bias or imprecision. Discuss both direction and magnitude of any potential bias
Author Response
Introduction
State specific objectives, including any pre-specified hypotheses.
Reply: The primary purpose of this study was to test our modified spectrophotometric assay for platelet aggregation. The secondary purpose was to validate this assay method to evaluate the quality of platelets contained in individual PRP preparations by establishing the reference range. Furthermore, our final goal was to add this assay method to the list of the point-of-care tests in PRP quality assurance and improve the quality of clinical evidence. These contents were summarized and inserted in the last paragraph of the Introduction section.
As for pre-specified or working hypotheses, if we could obtain relatively narrow range variations of the averages to clearly distinguish normal platelets from the dysfunctional ones, we hypothesize that our assay method could be applicable in testing platelet quality for our purpose.
Preparation of P-PRP and platelet suspension Give the eligibility criteria, and the sources and methods of selection of participants.
Reply: Participants were selected based on ineligibility. Those taking medication, especially anti-platelet drugs, and those with antibodies to specific viruses were excluded. According to laboratory reference ranges in healthy adults for blood tests, participants who showed “out-of-range” values in blood cell counts and in INR in prothrombin test were also excluded. The detailed eligibility criteria were added to a subsection in the Materials and Methods section.
The discussion should be modified Discuss limitations of the study, taking into account sources of potential bias or imprecision. Discuss both direction and magnitude of any potential bias.
Reply: Thank you for the comment. It is a little too difficult for us to list all the possible limitations and concisely discuss potential biases without causing a misunderstanding in the readers’ minds. However, we have tried.
There are several major limitations of this assay method, including biomedical, technical, and practical limitations in clinical setting. To avoid unnecessary extension, in this revision, we focused on the biomedical limitations.
This discussion was inserted in the last part of the Discussion section. We hope you will be satisfied with this revision.